# Research on the Impact of Heavy Rainfall Flooding on Urban Traffic Network Based on Road Topology: A Case Study of Xi'an City, China

Jiayu Liu [1,†] , Xiangyu Yang [2,†] and Shaobin Ren [1,*]

1    School of Architecture and Urban Planning, Huazhong University of Science and Technology,
     Wuhan 430074, China; m202274013@hust.edu.cn
2    School of Artificial Intelligence and Automation, Huazhong University of Science and Technology,
     Wuhan 430074, China; m202273255@hust.edu.cn
*    Correspondence: renhust@hust.edu.cn; Tel.: +86-157-3162-8859
†    These authors contributed equally to this work.

**Abstract:** In recent years, the frequent occurrence of extreme emergencies represented by heavy rainfall has posed a significant impact on urban transportation networks and caused great inconvenience to people's production and life. This paper examines the impact of heavy rainfall on the urban transportation network in Xi'an's main urban area, models the map of urban road network in the main urban area of Xi'an by applying complex network theory, quantifies and analyzes the topological and geographic structural characteristics of the affected sections (interrupted by heavy rainfall) due to heavy rainfall, as well as the impact of heavy rainfall on the functional indexes of the road network such as access efficiency and accessibility, and, finally, analyzes its formation causes and proposes targeted management countermeasures. The findings of this study serve as a valuable reference for enhancing urban traffic emergency management capabilities and fostering sustainable urban development.

**Keywords:** storm water flooding; urban transport; complex networks; topology; network efficiency

## 1. Introduction

Heavy rainfall is an extreme weather phenomenon characterized by high concentrations, heavy precipitation, and high intensity. In recent years, heavy rainfall and flooding disasters have occurred frequently in China, causing huge economic losses and even casualties. The urban transport system, as an important carrier of passenger and cargo transport and a vital part of daily life for residents, is extremely vulnerable to sudden events such as heavy rainfall [1]. Extreme weather conditions or unexpected events may cause multiple points of roadway failure, which can easily lead to a systemic breakdown of the transport network [2]. In other words, this can lead to traffic congestion, seriously affecting normal urban traffic and restricting traffic capacity badly.

As seen in Figure 1, flooding disasters are mainly located in the central and southern regions of the country and cause greater damage. The greatest hourly rainfall in the extraordinarily heavy rainstorm event that hit Zhengzhou in 2021 was 201.9 mm, breaking the record for the highest hourly rainfall in mainland China. The rainfall caused 37 breaks on trunk roads, over 200 breaks on rural roads, and the suspension of bus and metro services. In 2022, a total of 38 regional heavy rainfalls occurred in China, and 52 cities experienced 73 instances of flooding and waterlogging to varying degrees due to the heavy rain. These extreme rainfall events caused incalculable damage to urban transportation and seriously affected people's production and life. The prevention and control of urban flooding has become an important issue of urgent concern to society.

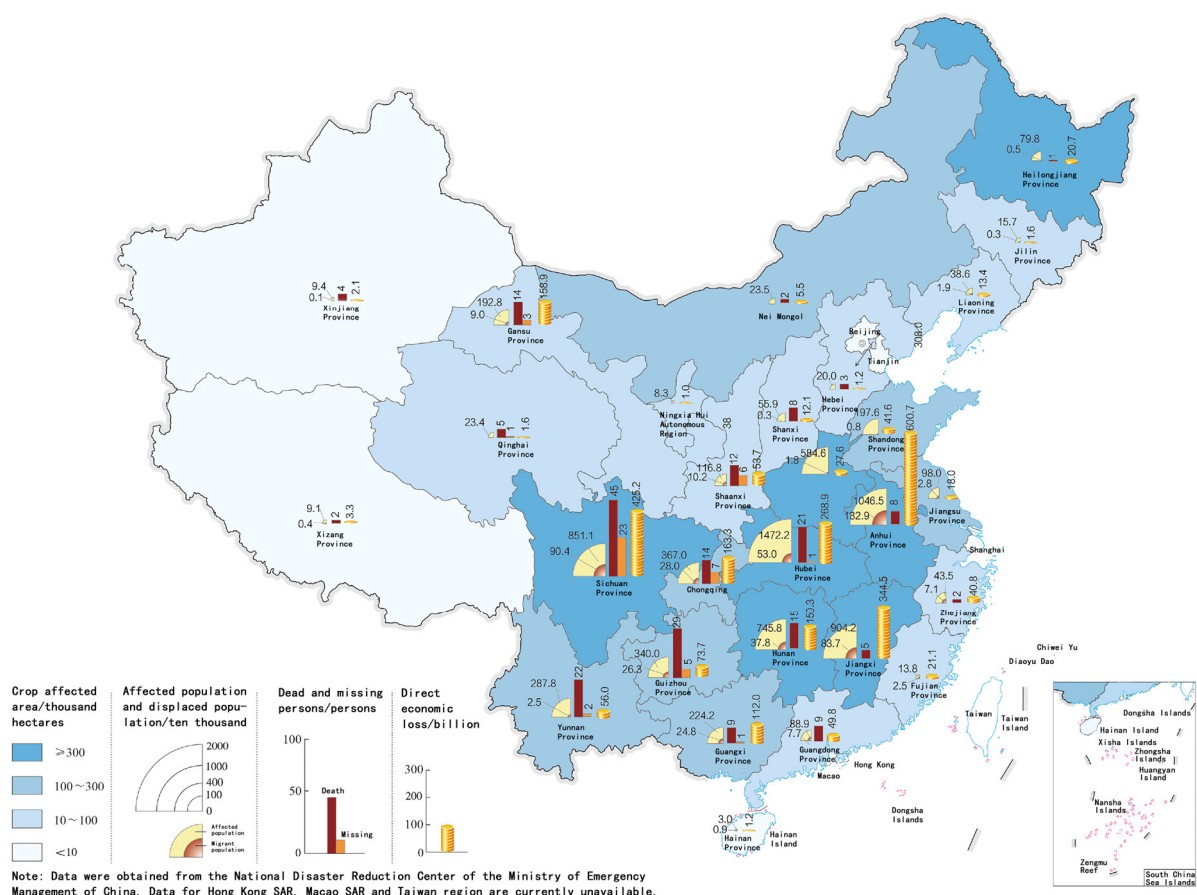

**Figure 1.** Flood distribution map of China in 2020.

In terms of the impact of heavy rainfall flooding on cities, current research has yielded fruitful results. For operational efficiency, Chenming Zhang et al. [1] used a spatial syntactic model to evaluate the changes in road network efficiency in Zhengzhou before and after the 7–20 storm disaster in terms of the "global-local" and "penetration-integration". Su et al. [2] used a vehicle-following model and a road traffic model (SUMO) to simulate traffic congestion on several main roads in Beijing on rainy days. For spatial distribution, Liyuan Zhao et al. [3] analyzed the impact of urban construction on the spatial distribution of storm waterlogging. Gori et al. [4] showed that land use change significantly increases the risk of flooding in urban waterfront areas by combining land use prediction and hydrological simulation methods.

Urban roads with many nodes are intricate. They can be seen as a complex network [5]. Translating a road network into a complex network and analyzing it on the basis of topology is an effective way to reveal the structural characteristics and operation of urban road networks [6]. Wei Zhenlin et al. [7] developed a composite transport network model to illustrate the impact of urban rail networks on the accessibility of urban transport networks in Beijing. Aura et al. [8] investigated the link between the vulnerability and accessibility of transport networks and clearly distinguished between these concepts. The topology of road networks from the perspective of complex networks is often studied in terms of betweenness [9], subgraph [10], and degree [11] as judgment indicators. However, the connotation of traffic network effectiveness is multifaceted. Multiple factors such as research objects, research perspectives, and premise assumptions can lead to differences in evaluation results. Therefore, there are still certain limitations in current research.

To sum up, most existing studies are based on spatial analysis software such as ArcGIS 10.2, and make subjective evaluations based on diagrams. They seldom consider quantitative analysis of the characteristics of the affected road sections and the impact of

rainstorms on traffic efficiency from the refined perspective of road topology. Given the above careful consideration, this study modeled the map of the urban road network in Xi'an's main urban area from the topological structure perspective based on the historical data. We analyzed the topological and geographic structural characteristics of the road sections damaged by unexpected events as well as the impact of heavy rainfall on the functional indicators of the road network. Then we analyzed the causes of their formation and provided optimization suggestions. It can be used for reference to improve the level of emergency management of urban traffic and the subsequent development of the city.

## 2. Materials and Methods

### 2.1. Study Area

Xi'an is the core city of the Guanzhong Plain City Cluster, the ninth national center city. It has a far-reaching role in the "three horizontal and three vertical" economic development system. Xi'an is located in the region with a temperate continental monsoon climate. It is hot and rainy in summer. The precipitation mainly concentrates between May and October, with typical aggregation [12]. According to the relevant departments, there are 306 major flood-prone spots in Shaanxi Province, of which 14 are in Xi'an. Furthermore, overpass tunnels and urban-rural interface areas are more prone to waterlogging areas. In this study, the main urban area of Xi'an is chosen as the study area (Figure 2). The main urban area is an alluvial plain, covering most of the flood risk area. In recent years, there has been an increasing trend of extremely heavy precipitation events, with frequent flooding disasters such as the "7–24" very heavy rainfall in 2016 (total rainfall exceeding "once in 100 years"), the "7–10", and the "7–30" very heavy rainfall in 2020 (high frequency, short interval). On 4 August 2022, Xi'an was hit by heavy rainfall with a maximum hourly rainfall intensity of over 50 mm, resulting in severe waterlogging and obstruction of traffic on some road sections. Among them, Yanta District has the largest number of disaster-hit roads. This emergency caused regional traffic paralysis, which brought great inconvenience to residents' traveling life and caused a large social impact. Therefore, the study is extremely important for Xi'an's control of urban flooding.

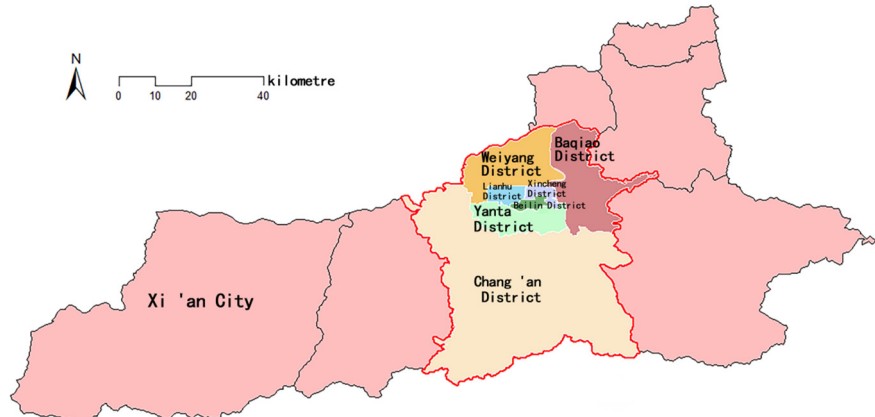

**Figure 2.** Map of Xi'an's main urban areas and administrative divisions.

### 2.2. Data and Processing

#### 2.2.1. Data Sources

The data used in this paper include urban traffic network data, geographic information data of the affected road sections in Xi'an Yanta district on 4 August 2022, a digital elevation model (DEM) with SRTM 90 m accuracy, and urban population distribution data with three arcsec accuracy under the WGS84 coordinate system in 2020. The urban traffic network data, including urban road intersection data and road centerline data, were sourced from the open source map data community OpenStreetMap (www.openstreetmap.org accessed on 5 May 2023), which were are used to model and analyze the urban road network structure.

The data on affected road sections in the city were obtained from the disaster information reports released by mainstream media in Xi'an (https://mp.weixin.qq.com/s/eTgysdHw-23K8LepEZ4JYA, accessed on 10 February 2023), which were used to identify the affected sections of the road network. The DEM data were obtained from the official website of SRTM (CGIAR-CSI SRTM—SRTM 90 m DEM Digital Elevation Database), which were used to analyze the elevation characteristics of the affected sections. The urban population distribution data were obtained from the Global Human Settlement Dataset (Global Human Settlement—download—European Commission (europa.eu)), which were used to assess the number of people affected by the affected sections.

2.2.2. Partial Data Processing Methods

1.    Urban road network data

This paper used the "continuous road centerline" as the basic unit to analyze the topological characteristics of the road network. The original road network data was downloaded from OpenStreetMap, read by Python, and then cleaned. We removed duplicate intersections and redundant clusters and retained the largest clusters. The urban road network data include the latitude and longitude of the nodes, the length of the road section, the maximum speed allowed for the road section, and other information.

2.    Data on affected road sections

After acquiring the data for the affected road sections, we used ArcGIS to match it to the road network data. Then, we verified the corresponding network edge of the affected road segments through comparison and exported the road network data containing the affected information (Figure 3). The road network was modeled using MATLAB, with the network edges being roads and the network nodes being junctions. On this basis, we analyzed the topological and geographical characteristics of the affected road sections and quantified the impact of urban flooding on the functional indicators of the road network.

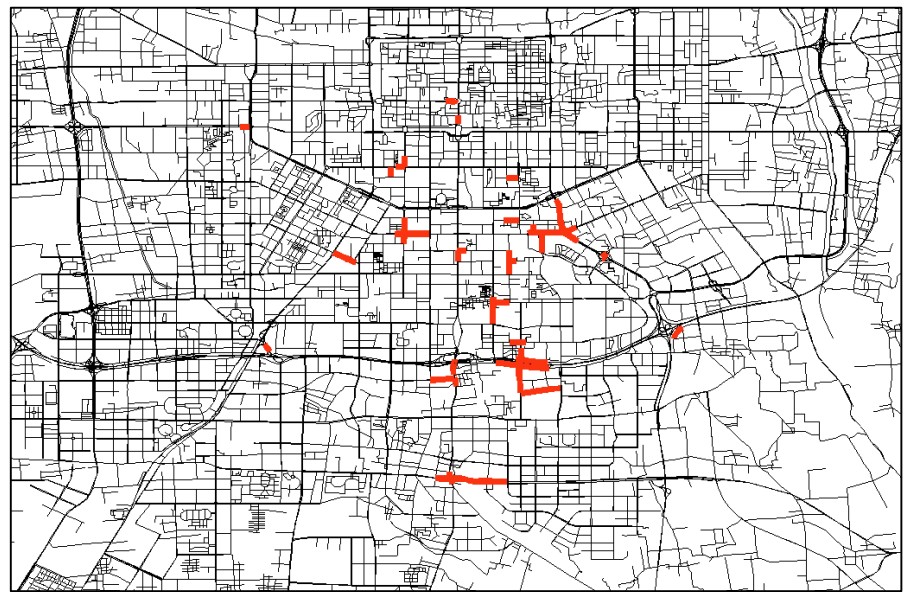

**Figure 3.** Marking of the affected section.

## 3. Construction of Urban Road Disaster Impact Indicators

*3.1. Indicators for Topological Analysis of Affected Sections*

3.1.1. Affected Section Nodes Importance

Urban road networks have scale-free characteristics [13]. Compared to random networks, scale-free networks without differential attacks are more fault-tolerant. However, when the network is attacked selectively, the scale-free network becomes more vulnerable.

According to studies, the system can crash if 5% of the core nodes are targeted preferentially [14]. In the case of the impact of heavy rainfall on urban traffic, the damage can be more severe if the core nodes are hit. Studies have mostly evaluated the significance of nodes in the road network using quantitative metrics [15]. In this study, we selected the node degree and the edge betweenness of affected roads for quantification.

1. Disaster node degree

The degree of a node *i* in an undirected network is the number of edges directly connected to the node, also known as the relational degree [16], and we define it as $D_i$. Degree has the advantage of being simple to calculate and intuitive to represent. The larger degree of a node indicates the higher centrality of the node, and the more important it is in the network. The degree of a node *i* is expressed as

$$D_i = \sum_{j \in \Gamma(i)} x_{ij} \tag{1}$$

where
$D_i$ is the degree of node *i*;
$\Gamma_i$ is the set of network nodes other than node *i*;
$x_{ij}$ is the accessibility of node *i* to node j in the network; if there is a directly connected edge between nodes *i* and *j*, then $x_{ij} = 1$, whereas $x_{ij} = 0$.
We calculated the average value of the node degree of the affected section and the average value of all node degrees, respectively, and compared them. As a result, we are able to gauge the significance of the impacted road within the network. The formula is as follows:

$$\overline{D_{aff}} = \frac{\sum\limits_{i \in \Gamma_{aff}} D_i}{N_{aff}} \tag{2}$$

where
$\overline{D_{aff}}$ is the average of the degrees of the nodes in the affected section;
$D_i$ is the degree of the node *i*;
$\Gamma_{aff}$ is the set of affected nodes;
$N_{aff}$ is the number of affected nodes.

2. The edge betweenness of the affected section [17]

The number of pathways passing through an edge divided by the total number of shortest paths in the network is known as the edge betweenness. We first calculated the minimum spanning tree of each node, then calculated the number of local shortest circuits by traversing the minimum spanning tree, and, finally, obtained the edge betweenness of each affected section. The larger the edge betweenness, the more the number of nodes passing through the road section in the shortest path, and the more critical the edge is in the whole road network. This can be expressed by the following equation:

$$B_{e_{lk}} = \frac{\sum\limits_{s=1}^{a} \sum\limits_{t=1}^{a} \frac{L(s,t,e_{lk})}{L(s,t)}}{a(a-1)} \tag{3}$$
$$(s \neq t, s \in a, t \in a, a > 1)$$

where
$e_{lk}$ is the connected edge in the network;
$B_{e_{lk}}$ is the edge betweenness connecting edge $e_{lk}$;
*a* is the total number of nodes in the network;
*s* and *t* are two different nodes in the network; we assumed that there are $L(s,t)$ shortest paths between nodes *s* and *t*, where there are $L(s,t,e_{lk})$ paths passing through the edge $e_{lk}$.

### 3.1.2. Topographic and Geomorphic Characteristics of the Affected Section

Generally speaking, road elevation and slope have a significant impact on rainwater storage. Elevation refers to the distance from a point along the plumb line to the absolute base, with higher road elevations indicating a greater vertical distance from the Yellow Sea level. The lowest location at a given spatial scale usually has a higher risk of flooding. The greater the cumulative elevation change, the more undulating the road; the smaller the cumulative elevation change, the flatter the road. The calculation formula is as follows:

$$E_{aff} = \sum \left| E_{nf} - E_t \right| \tag{4}$$

where

$E_{aff}$ is the cumulative elevation change of the affected road;

$E_{nf}$ and $E_{nt}$ represent the elevation of the nodes at each end of a road, respectively.

Slope refers to the degree of steepness of the ground, usually expressed as the ratio of the vertical distance to the horizontal distance of the slope (or slope ratio). In actual construction, roads are often set to a certain slope to facilitate drainage. China's current regulations require that the longitudinal slope of roads should not be less than 0.3%. However, if the slope of the ground is too large, the water potential will form an angle between the slope and the horizontal plane. The greater the slope, the greater the gravitational acceleration of the water flow, the faster the catchment rate, and the more prone to flooding at low points. The calculation formula is as follows:

$$i = \frac{h}{l} \times 100\% \tag{5}$$

where

$i$ is the slope of the affected section;

$h$ is the vertical distance of the affected road section;

$l$ is the horizontal distance of the affected road section.

### 3.1.3. Spatial Distribution Characteristics

1. Nearest neighbor index

The nearest neighbor index is the ratio of the actual nearest distance to the theoretical nearest distance. It is a geographical index that indicates the mutual proximity of point things in geographical space and can well reflect the spatial distribution pattern of disaster sites. The nearest neighbor distance of uniform distribution is the largest, random distribution is the second, and agglomerative distribution is the smallest. The calculation formula is as follows:

$$\overline{r_k} = \frac{1}{2\sqrt{n/A}} = \frac{1}{2\sqrt{D}} \tag{6}$$

$$R = \frac{\overline{r}}{r_k} = 2\sqrt{D}\overline{r} \tag{7}$$

where

$r_k$ is the theoretical nearest distance;

$A$ is the area of the study area;

$n$ is the number of affected nodes;

$D$ is the density of affected nodes in the area;

$\overline{r}$ is the average distance between the affected nodes and the nearest affected nodes;

$R$ is the nearest neighbor index; when $R < 1$, the waterlogging points tend to cluster; when $R = 1$, the waterlogging points are randomly distributed; when $R > 1$, the waterlogging points tend to be uniformly distributed.

2.     Clustering coefficient

The clustering coefficient is the ratio of the average distance between the node of the disaster section and the center of gravity of the affected section node, and the average distance between all nodes and the center of gravity of the affected section node in the road network. It describes the closeness between the node and other nodes. The larger the clustering coefficient, the lower the degree of clustering of the affected road section. The smaller the clustering coefficient, the higher the degree of clustering of the affected road section. It is calculated as follows:

$$C = \frac{d_{aff}}{d_{all}} \tag{8}$$

where

$C$ is the aggregation factor;

$d_{aff}$ is the average distance of the nodes of the affected section from the center of gravity of the affected section node;

$d_{all}$ is the average distance of all nodes from the center of gravity of the affected section node.

### 3.2. Indicators for the Impact of the Affected Road Sections on Traffic

### 3.2.1. Population Affected by the Disaster

The number of people affected in the disaster area can help us intuitively judge the impact of the emergency on the operation of the city. The larger the number of people affected and the larger the proportion of the total population within the scope of the study, the greater the impact degree of the disaster. The cumulative population directly affected by the road section can be expressed by the following formula:

$$P_{aff} = \sum_{i \in \Omega_{aff}} P_i \tag{9}$$

where

$P_{aff}$ is the total number of people affected by the storm;

$P_i$ is the number of people of the affected node $i$;

$\Omega_{aff}$ is the set of affected nodes.

### 3.2.2. Road Network Access Efficiency

The traffic efficiency of road network is a true reflection of the comprehensive operation condition of the traffic network, which reflects the traffic efficiency of passing through the road section and intersection in unit time. In this study, we used the average of the efficiency of all the nodes of the network to represent the overall network efficiency:

$$E = \frac{\sum\limits_{i \neq j} \frac{1}{t_{ij}}}{N(N-1)} \tag{10}$$

where

$E$ is the average value of node efficiency;

$t_{ij}$ is the minimum time spent from node $i$ to node $j$ in the network;

$N$ is the total number of nodes in the network.

### 3.2.3. Road Network Accessibility

Accessibility refers to the ease of getting from a given location to an activity using a specific transportation system [18]. Within a large region, the regional accessibility of each region refers to the ease of access of each region to various major centers of economic activity [19]. Road network accessibility measures the accessibility of opportunities from region $i$ to other regions [20]. This paper defined the number of opportunities as the corresponding population. The practical meaning of accessibility in this study is the

number of all populations in the network that can reach key node *i* at a given time, except for the key node.

The equation for accessibility based on the travel time threshold is as follows:

$$A_i = \sum_{j=1}^{n} D_i f\left(c_{ij}, T\right) \tag{11}$$

where

$D_j$ is the number of opportunities in region *j* (population size);
$c_{ij}$ is the cost of travel between region *i* and region *j*;
$T$ is the time threshold; when $c_{ij} < T$, the $f\left(c_{ij}, T\right) = 1$; otherwise, $f\left(c_{ij}, T\right) = 0$.

## 4. Results and Analysis

### 4.1. Topology of the Affected Section

#### 4.1.1. Importance of Road Sections

1.     Disaster node degree

Figure 4 shows that the degrees of nodes in the affected sections are all between 3 and 5. The median degree of all nodes is 3. The affected nodes are in the top 50% of all nodes, which means that they are all relatively significant. The average degree of the disaster node is 3.489, and the average degree of all nodes is 3.023. Again, it shows that the affected nodes have a larger degree relative to the overall road segment nodes, that is, the nodes of the affected road segment connect more edges and are more critical in the whole road system.

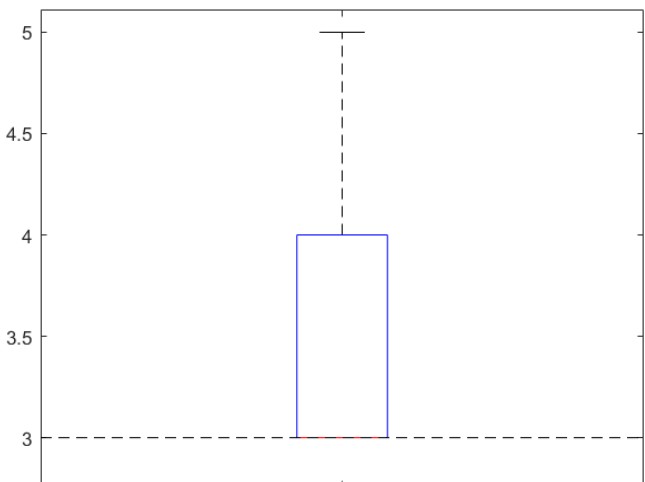

**Figure 4.** Distribution of node degrees on the affected sections.

2.     The edge betweenness of the affected section

Figure 5 shows the edge betweenness of the affected section. Figure 6 shows the relative values of the edge betweenness of the affected segments compared to the median of the edge betweenness of all segments. As can be seen, most of the relative values are greater than 1, indicating the relative importance of the affected sections. It also indicates that the affected section is the shortest route for most trips, and that the effect of a single point of impact will have a spillover effect, making the surrounding area less efficient overall. Among them, the section with the largest betweenness relative value (outlier) is the east side of the Old District Committee intersection in Chang'an District, with a relative value of 67.8702. This intersection is the center of people and traffic flow in Chang'an District, and plays an essential role in geography and culture. It is a vital central hub. Chang'an District has a long construction time, and the design capacity of the drainage pipe network is insufficient. In addition, due to the large terrain gap between East Chang'an

Avenue and West Chang'an Avenue, the flow kinetic energy is considerable in rainstorm weather. The rainwater pipe is more likely to be overloaded, making it prone to severe water accumulation.

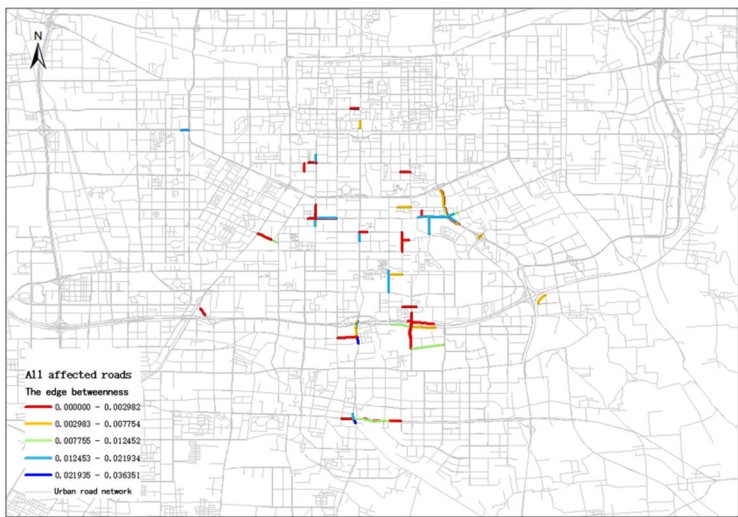

**Figure 5.** Number of affected sections.

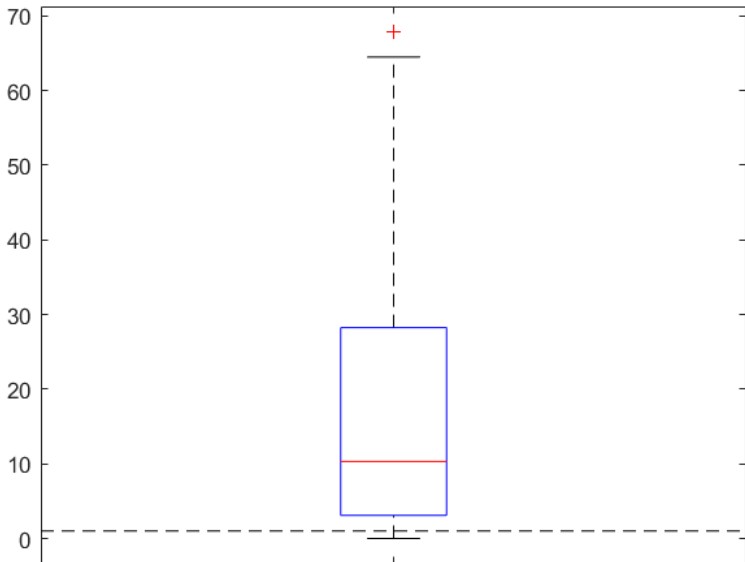

**Figure 6.** Relative values of the number of affected sections.

### 4.1.2. Topographic and Geomorphic Characteristics of the Affected Section

We used ArcGIS to process DEM data and obtained the elevation, slope, and other related parameters of Xi'an's main urban area.

Figure 7 shows the elevation characteristics of the main urban area of Xi'an. Generally speaking, the risk of flooding is lower at higher elevations; however, there is also a risk of flooding at higher elevations under extreme rainstorm conditions. By comparing the remote sensing image and the slope map, it was found that the overall elevation of the main urban area of Xi'an is low, and the slope is gentle, which makes the surface runoff movement slow. The areas with higher elevation are also prone to waterlogging under extreme rainfall.

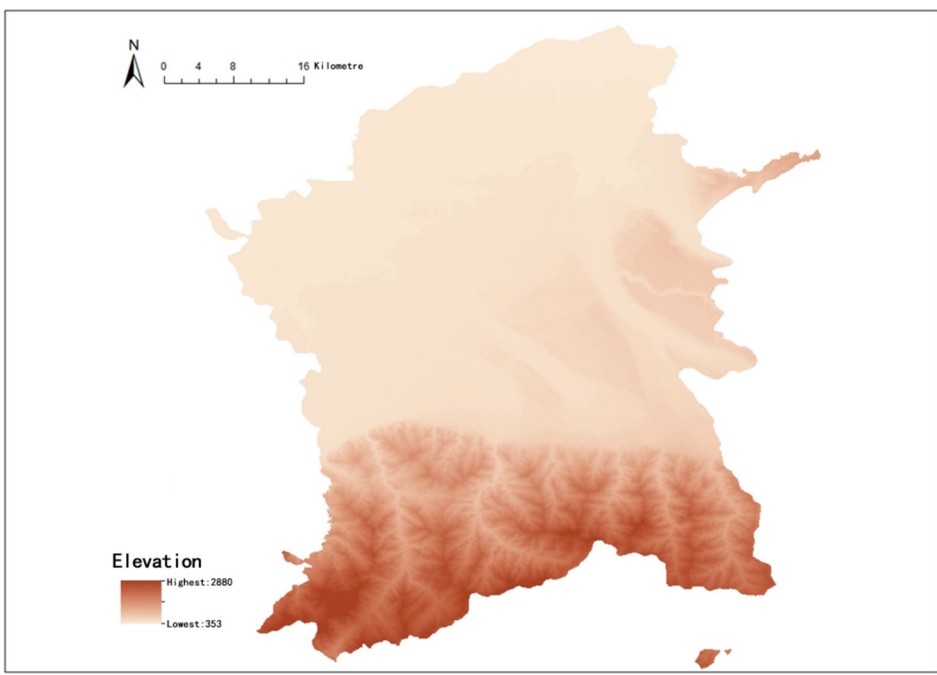

**Figure 7.** Elevation of the main city of Xi'an.

In the calculation of elevation difference, we first transformed the nodes at both ends of the affected road sections by latitude and longitude. Then we read out the elevation information of the two nodes separately and made the difference of the elevation values. At last, we summed up the absolute values of each to obtain the accumulated elevation difference. It is worth noting that Xi'an is located in the Guanzhong Plain area, the terrain is flat and there are almost no undulating road sections in the urban area [21] (although we do not exclude various undulating road sections, we believe their number is too small to have an impact on the results). The average difference in elevation of the affected sections is 4.625 m, while the average difference in elevation of all sections in the main city is 2.822 m. It is believed that the difference in elevation of the affected sections is much higher than the average difference in elevation of the city's sections. Water flow on the ground and underground pipelines accelerates to the affected nodes due to gravity, which is more likely to cause water accumulation.

In the calculation of slope, we calculated the horizontal distance between the two end nodes of the affected section according to the elevation difference and the length of the section, and then calculated the slope of the affected section using the formula. Figure 8 shows the slope values of each road section. The average slope of the affected section is 0.0175, while the average slope of all sections in the main urban area is 0.0147, indicating that the affected section has a larger slope and is more likely to cause water accumulation. (Supplementary statement: Because the nodes at the two ends of some short road sections are in two different grids divided by the elevation data set, the calculated elevation value is much higher than the normal value. According to the "Urban Road Engineering Design Code" CJJ 37-2012 (2016 edition) in China, the state stipulates that the slope of urban roads should not exceed 0.08. Therefore, we set the slope to 0.08 for any road over 0.08.)

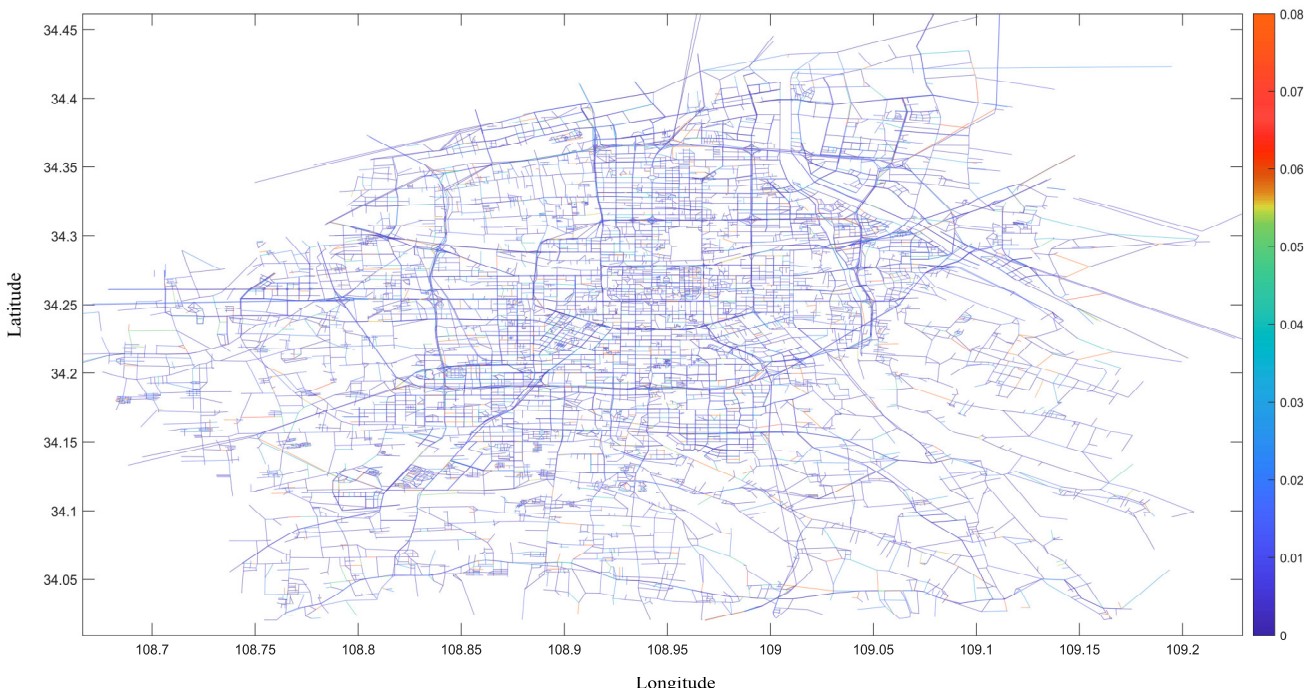

**Figure 8.** The slope value of each section in the main urban area of Xi'an.

4.1.3. Spatial Distribution Characteristics

1.    Nearest neighbor index

The actual closest proximity of the affected nodes is 122.2439 m. The theoretical closest proximity is 2151 m. The closest proximity index is 0.0568. This indicates that the actual closest proximity of the affected nodes is much smaller than the theoretical closest proximity. The distribution of the affected nodes is very compact.

2.    Clustering coefficient

The average aggregation coefficient of the affected nodes is 0.1305, indicating that the higher the degree of aggregation of the affected nodes. The analysis of the aggregation coefficient of the affected nodes shows that the affected nodes are densely distributed in the core urban area on the south side, with the largest number of nodes in Yanta District. In addition, the affected nodes are distributed along the main arterial roads, with waterlogging points mainly concentrated at important intersections, and entrances and exits of overpasses, but relatively few on branch roads.

*4.2. Impact on Traffic on Affected Roads*

4.2.1. Cumulative Population of Affected Roads

We first converted the longitude and latitude of each disaster section node, then put it into the global population data grid, and filtered out the nodes in the same grid to avoid repeated addition. Then, we summed up the population of the grid in which the node is located to calculate the number of people affected by the rainstorm disaster.

The rainstorm disaster affected the population in 41 grids, with a cumulative population of $3.69477 \times 105$ people. Compared with the number of $5.81526 \times 106$ permanent residents, it can be concluded that the proportion of the population affected by the rainstorm is about 6.35%. Then we rank the population number of the affected area from small to large, where the minimum number of affected people is 1176, and the maximum is 63,940. It can be seen that this rainstorm affects a massive amount of people and greatly impacts people's lives.

### 4.2.2. Network Access Efficiency

The overall network traffic efficiency was $2.96675 \times 10^{-2}$ before the disaster and $2.91941 \times 10^{-2}$ after the disaster, with a 1.6% decrease in network traffic efficiency. This indicates that the number of vehicles passing through road sections and junctions per unit of time has decreased and that the normal flow of traffic has been affected by the waterlogging of the storm. At the same time, the node failure of the affected section will make the traffic flow that should initially flow to the section flow to other sections, which may cause the flow of the surrounding section to exceed its tolerance, resulting in chain congestion of the entire urban section [1].

### 4.2.3. Accessibility Based on Time Thresholds

In this section, we selected the Xiaozhai intersection and the surrounding road sections as key nodes for analysis, with a degree of 4. Xiaozhai is located in the Yanta district of Xi'an, on the central axis of Xi'an, and its geographical location is very important. Xiaozhai area gathers a variety of functions such as commerce, schools, offices, hospitals, and tourism. It is the most active economic circle in Xi'an, with an average daily pedestrian flow of 500,000. Its marginal area is higher than the center area (basin). Therefore, its low-lying geographical feature makes it prone to internal flooding in heavy rainfall.

$c_{i,j}$ is the shortest time spent from point i to point j. For the affected road section, we increased the time cost to 106 orders of magnitude to simulate the road disconnection, and, finally, to calculate the reachability based on the time threshold. As the Xiaozhai area is very convenient and the study area is small, the 45 min travel cost can cover most of the study area. Therefore, in this study, the time thresholds were taken as T = 10 min, 15 min, 30 min, and 45 min. Table 1 presents the results of the calculations.

**Table 1.** Reachability based on time thresholds.

| Time Threshold T | Accessibility | Total Number of People Reachable | Disaster Accessibility | Total Number of People Affected | Percentage of Population Affected |
|---|---|---|---|---|---|
| 10 | 0.1638 | $9.5268 \times 10^5$ | 0.1517 | $8.8244 \times 10^5$ | 7.37% |
| 15 | 0.3917 | $2.2776 \times 10^6$ | 0.3754 | $2.1831 \times 10^6$ | 4.15% |
| 30 | 0.7990 | $4.6462 \times 10^6$ | 0.7895 | $4.5926 \times 10^6$ | 1.15% |
| 45 | 0.9336 | $5.4291 \times 10^7$ | 0.9241 | $5.3739 \times 10^7$ | 1.02% |

Comparing the data before and after the disaster, it can be seen that both the accessibility and the total number of people accessible are lower than before. This indicates that the cost of travel is further increased under the impact of heavy rainfall. Moreover, when the time domain value is 10 min, it affects 7.37% of the reachable population, indicating that the impact of rainstorm disasters is serious. At the same time, it further shows that the population distribution near the key nodes is more concentrated.

Figure 9 illustrates the distribution of reachable nodes for different time thresholds. As the time threshold increases, the number of reachable nodes increases, the reachability improves, and the number of people who can reach key nodes increases. Some nodes become unreachable within this threshold due to the rainstorm (Figure 10). Most of these nodes are concentrated in the periphery, but some adjacent nodes in the interior are also greatly affected. These nodes are mostly intersection nodes, which are more susceptible to traffic congestion. This is mainly due to their higher importance and being the point where many of the shortest paths pass through, with greater traffic flow.

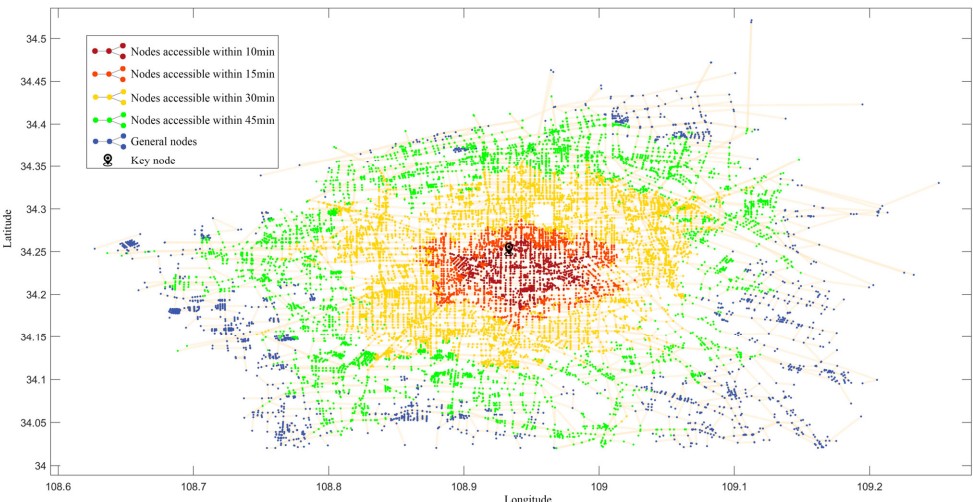

**Figure 9.** Reachable nodes in the network at different time thresholds.

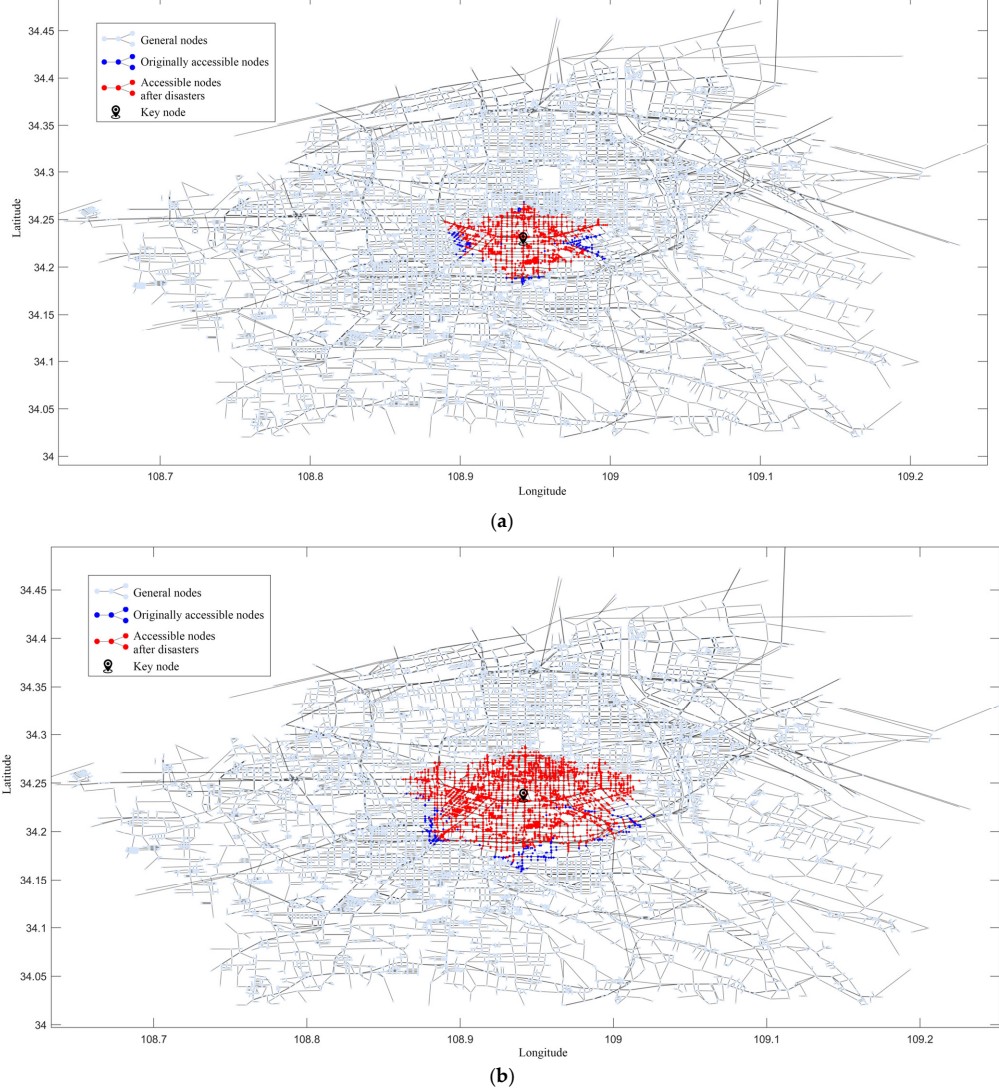

**Figure 10.** *Cont.*

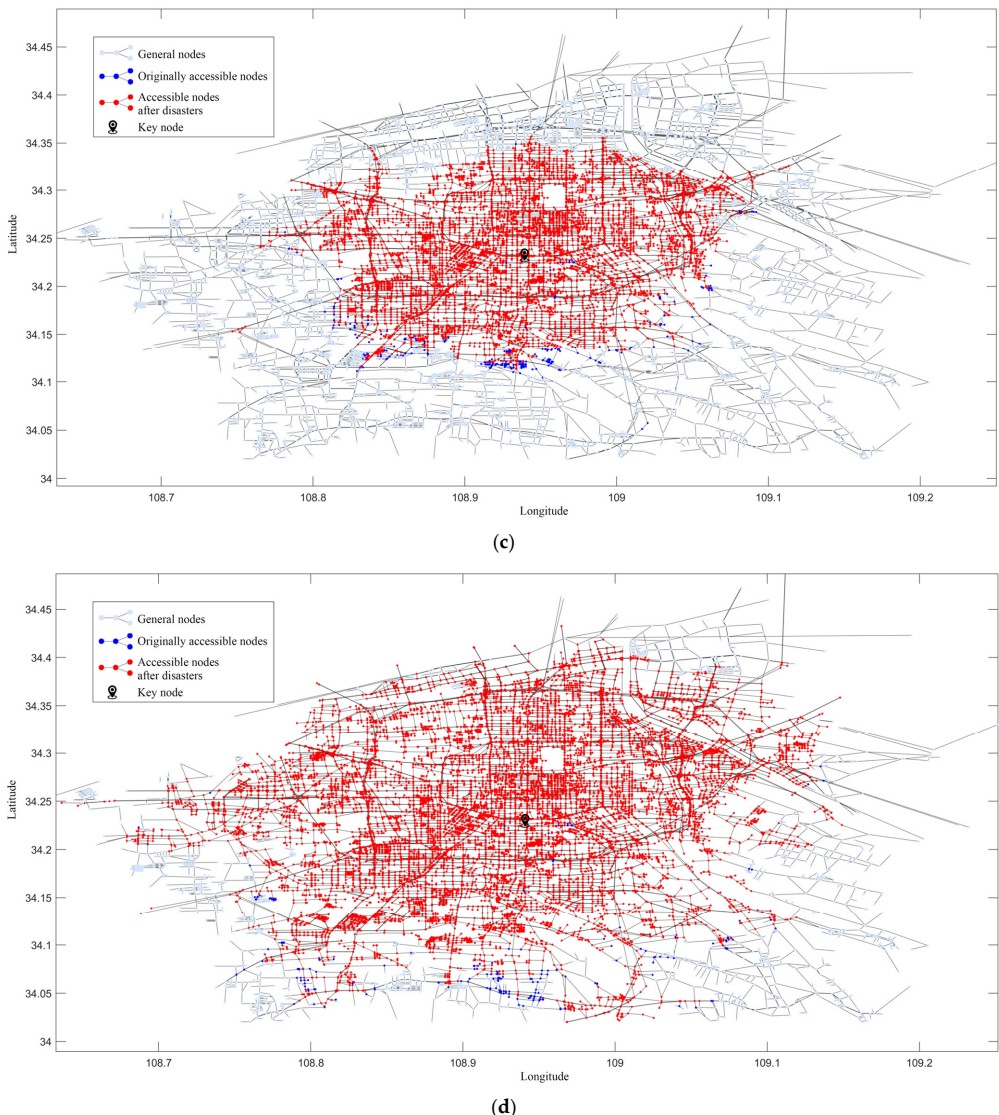

**Figure 10.** Distribution of reachable nodes before and after the disaster for different time thresholds. (**a**) T = 10 min; (**b**) T = 15 min; (**c**) T = 30 min; (**d**) T = 45 min.

## 5. Occurrence Mechanisms and Governance Recommendations

### 5.1. Causes of Waterlogging

#### 5.1.1. The Mismatch between the Construction Level of Water Conservancy Facilities and Transportation Planning

The above research shows that waterlogged nodes tend to be more important in the overall urban road system, connecting more edges. Additionally, they are more greatly distributed in the entrance and exit of expressways and trunk roads, carrying more traffic functions. However, the water conservancy facilities near such important nodes are often not strengthened, and some of them are even aging and damaged due to high use intensity. On the one hand, urban construction attaches more importance to the overground and belittles the underground. The construction standard of underground pipeline networks is low, with little investment and low maintenance level. In Xi'an, the construction level of drainage facilities and rainwater and sewage networks is generally low, primarily 1 to 3a. The standard for crucial areas such as railway stations and storage sites is only 3a, and the construction level of rain and pollution pipe networks in transportation hubs and important road nodes has not been strengthened. In the event of heavy rainstorms with high recurrence periods, the drainage capacity of the city's drainage system will not meet

the drainage demand [22]. This situation is common throughout China. According to the surveys, more than 70% of urban drainage projects have a recurrence interval of less than one year, and in major parts of old urban areas, 90% are even below the lower limit specified in the code.

On the other hand, the drainage capacity of the pipe network in some areas, especially in old urban areas, is weak, with insufficient self-drainage and pumping capacity. The phenomenon of mixed and misconnected pipes is serious. In this heavy rainstorm in Xi'an, a number of nodes in the old city were seriously affected. Xi'an has about 112.5 km of combined rainwater pipes with an average drainage pipe compliance rate of only 42.18%, has 11 drainage pumping stations with a compliance rate of only 27%, and the distribution density is high in the newly built areas. In addition, the capacity of the regulating ponds and the diameter of the inlet pipes are small, which has caused some of the drainage pipes to be blocked. Coupled with the inability to clean them in time, it has resulted in the combined flow of rainwater and sewage, easily causing the phenomenon of siltation in the pipe network which then affects its drainage capacity.

### 5.1.2. Inadequate Road Detail Design

The pavement design is insufficient and the detailed data on such factors as road elevation and longitudinal section slope have not been considered. When determining the road elevation, the actual terrain of the site was not taken into account, and the design of underground pipelines, ground drainage, and buildings on both sides were not considered comprehensively. Secondly, due to the long history of Xi'an, modern urban construction usually broadens and improves roads in accordance with the original historical grid roads without considering the planning and development of the surrounding areas of the roads, which makes the drainage system unable to accommodate the increased rainwater runoff due to urban development. Most waterlogging prone points are distributed in intersections and tunnels in built-up areas.

The paving materials of the road are mostly cement concrete and other impervious materials, and the surface runoff increases. With the rapid economic development of Xi'an, the scale of the built-up areas of cities and towns has expanded rapidly, with a large amount of non-construction land, such as forest land, arable land, and water areas being transformed into construction land. The original permeable ground is gradually replaced by impermeable surfaces such as cement concrete. In 2022, the urban built-up area of Xi'an was estimated to be 680 km$^2$, over seven times that of 1978. The gradual increase in impervious areas within the city has led to a decrease in infiltration and an increase in surface runoff. The faster rate of rainwater confluence has led to the lack of storage processes. At the same time, in order to meet the needs of vehicle travel, urban road design emphasizes "flat, straight and wide". When road construction is carried out, the terrain changes, many green vegetation is destroyed, and the original ecological environment is disturbed. The imbalance of ecological environment makes the city's ability to self-regulate rainwater greatly reduced and further increases the risk of rainstorms waterlogging in the city, endangering the life and property safety of residents.

### 5.1.3. Special Climatic and Construction Conditions

Abnormal climate is the direct and common cause of waterlogging. The enormous emissions of $CO_2$ have led to the "heat island effect" in the city, and climate change such as global warming has caused an unstable atmospheric structure and frequent convective precipitation extremes. Statistics show that the overall precipitation trend in Xi'an has been decreasing in recent years, with a rate of decrease of 6.05 mm/10a. However, the frequency of extreme rainfall events has been increasing yearly, with an inclination of 0.28 (mm/d)/10a [23]. It has shown a more pronounced increase in the central city than in the surrounding districts and counties. Nevertheless, the frequency and intensity of heavy precipitation as flooding-causing factors contribute less to urban flooding than other factors.

This is different from the southeast coastal cities, and also reflects the unique flooding characteristics in the semi-arid region of northwest China.

At the same time, Xi'an is short of water resources. Years of groundwater exploitation not only caused a large decrease of urban groundwater level, but also caused a series of environmental geological problems such as ground subsidence and ground cracks in urban areas. Xi'an's five historical lakes have also lost their storage functions to a great extent. A series of climate, environmental, and resource issues have led to frequent flooding in the city.

*5.2. Governance Recommendations*

There are many causes of stormwater flooding, so it is important to identify the key factors and take targeted measures. This paper put forward some governance suggestions mainly from "prediction, solution and enhancement".

Firstly, we should carry out forecasting and early warning work on flooding. The government can set up platforms for forecasting and warning, using numerical models of rainfall and flooding processes to carry out forecasting. At the same time, we should realize "smart flood control" by synchronizing flood information and risk data to the decision support system of relevant government departments, such as vehicle navigation, urban management, and traffic command [24].

Secondly, we should strengthen the vertical design of urban roads. In terms of infrastructure, an urban drainage and waterlogging prevention engineering system of "source emission reduction, pipe network discharge, combination of storage and drainage, and emergency exceeding the standard" has been built, the drainage capacity of pumping pump stations in areas prone to flood accumulation and waterlogging has been improved, and the rain and pollution diversion transformation of the urban drainage pipe network has been carried out. According to the types of nodes affected by the disaster, the countermeasures of "promoting classified policies and tapping the stock potential" are adopted, and the corresponding solutions are proposed according to the city's traffic road distribution, travel demand, and other factors, combined with the internal causes of congestion in different types of areas. In the vertical design of roads, relevant personnel should fully analyze the site conditions before construction, make reasonable use of landforms, control the elevation and slope of roads and major infrastructure, and strengthen the research on the influence and method of vertical elevation control to better guide the subsequent design. Strengthen the connection of flood control and drainage planning, drainage planning, comprehensive pipeline planning, and other related planning, and properly handle the relationship between roads, buildings, and landscapes.

Finally, speed up the construction of sponge cities with grey-green facilities. Build or renovate green roofs and permeable roads to increase the urban permeable area, avoid surface runoff, and make rainwater infiltrate into the ground from the source. Utilize natural terrain, geomorphology, or micro-terrain adjustment to disperse the rain and avoid water. Construct rain gardens and grass planting ditches to strengthen the capacity of urban rainwater retention, in order to regulate and stagger peaks. Purify the collected rainwater through sewage treatment facilities and ecological levees and re-apply it to greening irrigation and fire fighting. This is of great significance for water resource utilization and water environment protection and can improve the ecological environment. At the same time, take engineering measures to strengthen the construction of grey facilities in areas with serious waterlogging, such as building underground comprehensive pipe galleries to improve drainage capacity.

## 6. Summary

The serious impact of urban flooding has become a pressing problem everywhere. This study used the main urban area of Xi'an as empirical evidence to quantify the impact of heavy rainfall disasters on urban transport networks by building a road topology model. The main findings are as follows:

The degree of the affected nodes and the betweenness of the affected road sections are both greater than the median of the overall road network, indicating the high importance of the affected road sections. The generally high elevation of the affected road sections and the large elevation difference indicate that the higher elevations are also at great risk of flooding under extreme rainfall conditions. The road gradient is a key factor in determining whether or not a road is affected. The nodes affected were compactly distributed and concentrated around the main road sections.

The rainstorm affected a large population and reduced the efficiency of the road network, causing a large impact on people's lives and travel. This study took "Xiaozhai" as the key node to conduct the accessibility analysis based on time threshold and carried out a quantitative study on the impact of rainstorm on urban traffic.

Combining with the analysis of the actual situation of Xi'an city, this study concluded three main reasons for waterlogging: the mismatch between the construction level of water conservancy facilities and transportation planning, inadequate road detail design, and the influence of extreme weather. Subsequently, this study put forward differentiated solution strategies for different problems.

This study quantitatively described the multiple features of the affected nodes and sections and described the impact degree of the rainstorm disaster on the urban traffic network, which is scientific and universal. When a city suffers from various kinds of meteorological or other disasters and faces the risk of traffic stagnation, this method can be applied in risk assessment, identification of key traffic nodes, and improvement of urban emergency management ability. It should be noted that this paper only considers the mode of travel by car to quantify transport indicators and does not consider complex public transport systems such as the metro, which can be further developed in subsequent studies. However, this paper's evaluation indicators and quantification methods can still provide decision support for urban disaster risk management prevention and sustainable urban development.

**Author Contributions:** Conceptualization, J.L. and X.Y.; methodology, J.L. and X.Y.; software, X.Y.; validation, J.L., X.Y. and S.R.; formal analysis, X.Y.; investigation, X.Y.; resources, X.Y.; data curation, X.Y.; writing—original draft preparation, J.L.; writing—review and editing, S.R.; visualization, J.L.; supervision, S.R.; funding acquisition, S.R. All authors have read and agreed to the published version of the manuscript.

**Funding:** This research was funded by The Humanities and Social Sciences Program of the Ministry of Education in China, grant number 19YJAZH075.

**Data Availability Statement:** The data presented in this study are available on request from the author.

**Conflicts of Interest:** The authors declare no conflict of interest.

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
