# Peer review of "Research on the Impact of Heavy Rainfall Flooding on Urban Traffic Network Based on Road Topology: A Case Study of Xi’an City, China"

_land, doi:10.3390/land12071355_

Round 1
Reviewer 1 Report
Very relevant topic and good work. I recommend for publication.
However, some observations from my end.
1. Line 155-56mention edge betweenness.
2. Line 200, why Eaff is not calculated per km, and what will happen to this cumulative terrain if the terrain is rolling in nature (wave like).
3. Line 237 - what will happen if a node is not affected but its surrounding nodes/edges are which makes it isolated in the network; This is going to be underestimation; rather it is better to find out if portions of network that will be isolated and then find the population in that.
4. Line 265 - Is population of a sub-regions within a city region a good indicator of opportunities - most of the employment zones contain less population and vice versa. What does this city statistics tell?
5. Line 301 - Section 4.1.2, why not treat elevation and slope separately? And Line 322-23 where it is said that undulating sections are affected more requires further clarification. Do they create concave or convex regions i.e. bowl shaped land forms?
6. Line 329 - Please give explanation why nodes located closely are affected more. Are they part of dense urban development where drainage system is inadequate or fails often?
7. Line 387 - portions missing, I think.
8. Line 417 - water damage, water construction...are these words commonly used in Government Technical Literature...or else change them, if possible.
9. Line 445 - Would it be Cement Concrete?
10. Line 488 - What does "strengthen vertical design" mean, please explain.
11. Line 495 - If core area is most affected, how is "sponge city" going to help or even be possible. Please hint on its feasibility.
Author Response
Dear reviewer,
Thank you very much for your comments and professional advice on our manuscript entitled “Research on the Impact of Heavy Rainfall Flooding on Urban Traffic Network Based on Road Topology: a case study of Xi'an City, China”. We are very grateful to receive your comments, which are very helpful for improving our paper. We have sincerely and carefully considered the comments and revised the manuscript thoroughly. Detailed point by point response is given below. Revised portions are highlighted in RED in the revised manuscript.
I hope that the revised manuscript is suitable for publication in Land. Thank you again for your help with the manuscript. I am more than happy to provide any further information you may need.
Point 1: Line 155-56 mention edge betweenness.
Response 1: We sincerely thank the reviewer for careful reading. As suggested by the reviewer, we have corrected the “betweenness” to “edge betweenness”.
Point 2 : Line 200, why Eaff is not calculated per km, and what will happen to this cumulative terrain if the terrain is rolling in nature (wave like).
Response 2 : Thank you for your professional advice. Your approach does help to avoid some special cases, which we did ignore in our experiments. Unfortunately, due to the adaptability of the technical methods in this paper and the actual terrain conditions, it is difficult for us to calculate in kilometers. The reasons are as follows:
First, we provided a complementary argument on the experimental data. In the road network data, we used the "node-edge" form of data structure to represent the road segment, which was defined by two end points: the start node indicating the start position of the road segment, and the end node indicating the end position of the road segment. A total of 41699 road sections are involved in this experiment, of which 2024 are longer than 1 km and 451 are longer than 2 km. That is, only 4.85% of the roads are larger than one kilometer, and 1.08% of the roads are larger than two kilometers. Calculating in kilometers doesn't make sense for most road segments. At the same time, the calculation of other parameters in this experiment is based on the road segment as the basic unit. If we also take the road section as the basic unit for the calculation of elevation, the measurement values in the experiment can be unified.
Secondly, according to the industry standard "Urban Road Engineering Design Code" (CJJ37-2012) issued by the Ministry of Housing and Urban-Rural Development of China, in a section of road, the elevation of the main node should be determined first and set in sections, and the slope of the road should be as close as possible within a certain range. Therefore, if urban roads are built according to the specifications, wavy terrain will not appear over short distances.
Finally, the research object we selected in this experiment is the main urban area of Xi'an City, which is located in the Guanzhong Plain and has a flat terrain. There are almost no wavy road sections in the urban area (There may be still a few wavy segments, but we think the number is too small and we can ignore them). You can also see from fig.8 The slope value of each section in the main urban area of Xi'an in line 357 that the slope in many sections is close to 0. Similarly, Gao Shan and Sun Hu (2022) calculated the surface relief degree of Xi'an by grid technology, which proved that the surface relief value of Xi'an was low. We also illustrated this feature of Xi'an city in Section 4.1.2 of the manuscript.
References:
Shan Gao, Hu Sun, Yinyao Xu, and Shubo Sun Evaluation on Suitability for Human Settlements in Xi’ an Based on Grid (in Chinese). Journal of Ningxia University(Natural Science Edition),2022,43(01):115-120.
Point 3 : Line 237 - what will happen if a node is not affected but its surrounding nodes/edges are which makes it isolated in the network; This is going to be underestimation; rather it is better to find out if portions of network that will be isolated and then find the population in that.
Response 3 : Thank you for your professional advice. It helps us a lot. Nodes in this article represent traffic nodes, such as intersections. If the nodes connected to this node are all disaster nodes, it means that the node has almost completely lost the function of traffic peers, and the population located at this node may have difficulty in reaching other nodes by means of transportation. In this study, population data were derived from GHSL-Global Human Settlement Layer. The dataset divides the statistical region into several 0.0083*0.0083 grids. We determined the population affected by the rainstorm by counting the population of the grid where the disaster node is located. In this paper, there are 131 affected nodes, but only the population in 41 grids is affected. There are 23 affected nodes isolated, distributed in 9 cells. It affected 1.29 x 105 people.
Point 4 : Line 265 - Is population of a sub-regions within a city region a good indicator of opportunities - most of the employment zones contain less population and vice versa. What does this city statistics tell?
Response 4 : Thank you for your professional advice. It is true that most employment areas have a small population, but in this experiment, the area we study is the number of opportunities for each area within the main urban area to reach the key node, which covers all types of land, including commercial land, residential land, and many other types. This will not be limited to employment zones. Therefore, we believe that the calculation results of this experiment are not affected by land use properties.
As early as 1979, Morris et al. [1] gave a classic definition of accessibility. They believed that accessibility refers to the convenience of using a specific transportation system to reach an activity place from a given location. Dadao Lu [2] believes that the regional accessibility of each region within a large regional scope refers to the convenience of each region (starting from its economic center) to various major economic activity centers. From the perspective of the spatial configuration of the network, most existing studies measure the accessibility of a certain point by the convenience of people flow, logistics and information flow [3]. In this experiment, we use population as an indicator to quantify "accessibility", that is, the accessibility of a certain point is measured by the convenience of people flow. For consumers arriving at commercial zones, the accessibility of commercial areas is defined as the extent to which land use and transportation systems can enable individuals (or groups) to reach a certain commercial place (or destination) within a certain time range through a mode of transportation (or a group of modes of transportation).
Many scholars use population size as an indicator to measure accessibility. For example, Miller [4] formulated space-time and explored the effect of geographical context and scale on individual accessibility. Kwan [5] divided accessibility into individual accessibility, which is a good indicator of the life quality of individual cattle, and local accessibility, which refers to the attributes unique to locations or places easily accessible to all populations, that is, the ability of a location to be "approached". Zhixue Zhang and Tongsheng Li [6] used the non-agricultural population of economic activity centers to quantify the accessibility of regional nodes.
References:
- Morris J M, Dumble P L, Wigan M R. Accessibility indicators for transport planning. Transportation Research A, 1978,13(2):91- 109.
- Dadao Chen.Regional development and its spatial structure.Beijing: Science Press, 1995.117-124.
- Chen, J., F. Lu, and C. X. Cheng. "Advance in accessibility evaluation approaches and applications." Progress in Geography5 (2007): 100-110.
- Miller, Harvey J. "Measuring space‐time accessibility benefits within transportation networks: Basic theory and computational procedures." Geographical analysis1 (1999): 187-212.
- Kwan M P,Muray A T. Recent advances in accessibility researceh: Representation,methodology and applications . Geo-graphical Systems,2003,5: 129~138.
- Zhixue, Zhang, and Li Tongsheng. "Study on GIS-Based County-Level Transport Accessibility: A Case Study of Shaanxi Province." Human Geography1 (2010): 100-104.
Point 5 : Line 301 - Section 4.1.2, why not treat elevation and slope separately? And Line 322-23 where it is said that undulating sections are affected more requires further clarification. Do they create concave or convex regions i.e. bowl shaped land forms?
Response 5 : Thanks for your professional suggestion. We have treat elevation and slope separately in section 3.1.2 and section 4,1,2. Besides, the original description of surface undulation was controversial, and we replaced it with a description of the slope of the road to make it more clear.
Point 6 : Line 329 - Please give explanation why nodes located closely are affected more. Are they part of dense urban development where drainage system is inadequate or fails often?
Response 6 : Thank you very much for your advice. We didn't think it through properly in this regard. In this section, through data analysis, we found that the distribution of affected nodes in this rainstorm disaster event is relatively compact, and the spatial characteristics of their distribution are revealed. However, I am sorry that this conclusion may not apply to all cities. Due to the differences of various factors such as the construction degree of different cities, geography and climate, their disaster-prone nodes will show their laws.
As for the compact spatial distribution of waterlogging points in Xi 'an, we believe that it is caused by Xi 'an's circular expansion of urban structure, which strengthens the "focus" of social activities on the urban core in the process of urbanization, and also increases the burden of central nodes and the vulnerability of nodes. This may require some data to further demonstrate, and we will continue to deepen relevant research.
Point 7 : Line 387 - portions missing, I think.
Response 7 : We were really sorry for our careless mistakes. Thank you for your reminder. I have added the word “Comparing” in the manuscript.
Point 8 : Line 417 - water damage, water construction...are these words commonly used in Government Technical Literature...or else change them, if possible.
Response 8 : Thanks for your professional suggestion. After consulting a large number of reference materials, we have changed these phrases.
Point 9 : Line 445 - Would it be Cement Concrete?
Response 9 : Thanks for your careful checks. We are sorry for our carelessness. Based on your comments, we have made the corrections to make the word more specific.
Point 10 : Line 488 - What does "strengthen vertical design" mean, please explain.
Response 10 : We think this is an excellent suggestion. We have explained "strengthen vertical design", and you can find it in the updated manuscript.
Point 11 : Line 495 - If core area is most affected, how is "sponge city" going to help or even be possible. Please hint on its feasibility.
Response 11 : Thank you for your suggestions. We think this is an excellent suggestion. Now I will explain sponge cities and their role:
Sponge City is a scientific urban water management strategy proposed by Chinese architect Professor Kongjian Yu based on China's national conditions. Its concept has undergone rigorous scientific demonstration, which is also consistent with the international concept of urban rainwater management. It mainly refers to the "infiltration, stagnation, storage, purification, use, discharge" and other technical ways to achieve a urban benign hydrological cycle, improve the infiltration, storage, purification, utilization, and discharge capacity of runoff rainwater, and maintain or restore the sponge function of the city. It has a great prospect in the comprehensive realization of urban flood control and waterlogging, water resources utilization, water environment protection and water ecological restoration. Many Chinese scholars have also demonstrated its effect: Huang, et al., [1] used a two-dimensional hydrodynamic model to analyze the waterlogging reduction effect of sponge city construction pilot, and concluded that sponge city construction can increase the infiltration rate of the underlying surface, the depth of surface water accumulation, the amount of water accumulation, and the area of water accumulation. Sha chen, and Xiaohong Chen [2] took Foshan City as an example to analyze the reduction effect of sponge city, and the maximum reduction rate can reach nearly 80%. Therefore, the sponge city can help alleviate the flood disaster in the core area through its multiple working principles. The specific practices of sponge cities have been added in the manuscript, which is shown in line
But as mentioned in the second line of section 5.2, we just chose Sponge cities as a way to enhance stormwater management. Whether the concept is proposed or the official document is explained, sponge cities are not considered as the main means to solve the urban flooding problem. Some measures of sponge city construction can improve the waterlogging situation caused by insufficient urban drainage systems to a certain extent, especially the waterlogging caused by poor drainage in local plots (such as communities, etc.), but it cannot avoid the occurrence of city-wide waterlogging. Only through rivers, lakes, wetlands and other regulations and expansion of drainage pipe network and drainage pumping stations can fundamentally solve the problem of large-scale urban waterlogging [3] [4] .
References:
- Miansong Huang, et al. "Numerical simulation of urban waterlogging reduction effect in Guyuan sponge city." Water Resources Protection5 (2019): 13-18.
- Sha chen, and Xiaohong Chen. "Simulation of urban rainfall runoff pollution and control effect by low impact development." Water Resources Protection5 (2018): 13-19.
- Zongxue Xu, Tao Cheng, and Meifang Ren, "When should "City look at the sea" stop -- discuss the function and role of sponge city" (in Chinese), China Flood and Drought Management,2017,27(05):64-66+95.
- Qiting Zuo. "Water science issues in sponge city construction."Water Resources Protection4 (2016): 21-26.
Thankyou very much for your attention and time. Look forward to hearing from you.
Yours sincerely ,
Jiayu Liu, Xiangyu Yang, and Shaobin Ren
29 June 2023

Reviewer 2 Report
The paper is well-structured and presents a comprehensive examination of the impact of heavy rainfall flooding on urban traffic networks. The authors have conducted significant experimentation to support their arguments. However, there are several areas that require attention to enhance clarity and strengthen the paper's contributions. The following points address these concerns:
1) The title indicates that the study is based on road topology, implying the utilization of complex network theory and topological characteristics of the road network. However, the approach employed in this study appears to generate a common topological network, which is commonly used in road network production, particularly in GIS approaches. It is essential to clarify how this paper contributes to the understanding of road topology and what distinguishes the approach used in this study from previous research. Providing these details will enhance the paper's overall contribution.
2) The paper lists three causes of water damage: 1) lagging construction of water conservation facilities, 2) increased percentage of impervious area, and 3) special climatic and construction conditions. However, it is unclear how these causes were detected and assessed in this study. Are they assumptions made by the authors or based on statistical data from other research? Linking these causes back to the results of the current study will strengthen the arguments and provide a clearer understanding of their relevance.
3) In section 5.2, the paper presents general governance recommendations. To improve the practicality and effectiveness of these recommendations, it would be beneficial to link them back to the specific results and findings of the study. This linkage will demonstrate how the proposed recommendations address the identified issues and align with the study's outcomes, enhancing the paper's overall impact.
4) The paper would benefit from a clearer statement regarding the main contribution of this research. It is important to explicitly state the unique contributions that this study offers to the field. This clarification will enable readers to better understand the significance and value of the research presented in the paper.
5) The abstract is generally well-written. However, a few minor improvements could enhance its clarity and readability. Here are a few suggestions:
a) Consider rephrasing the sentence "This paper takes the main urban area of Xi'an as the research object" to "This paper focuses on studying the main urban area of Xi'an" or "This paper examines the impact of heavy rainfall on the urban transportation network in Xi'an's main urban area." This helps avoid the use of "research object," which may sound a bit technical.
b) In the sentence "quantifies and analyzes the topological and geographic structural characteristics of the affected sections due to heavy rainfall," consider clarifying what is meant by "affected sections." Are these sections of the road network that experience damage or disruptions due to heavy rainfall? Providing a brief explanation will improve understanding.
c) The phrase "impact of heavy rainfall on the functional indexes of the road network" can be made more specific. What specific functional indexes are being considered? For example, you could mention parameters like traffic flow, congestion, travel time, or accessibility.
d) Consider restructuring the sentence "It is an important reference for improving urban traffic emergency management capability and promoting sustainable urban development." It could be revised as follows: "The findings of this study serve as a valuable reference for enhancing urban traffic emergency management capabilities and fostering sustainable urban development."
Author Response
Dear reviewer,
We feel great thanks for your professional review work on our article. As you are concerned, there are several problems that need to be addressed. According to your nice suggestions, we have carefully revised the whole manuscript, the detailed corrections are listed below.
I hope that the revised manuscript is suitable for publication in Land. Thank you again for your help with the manuscript. I am more than happy to provide any further information you may need.
Point 1 : The title indicates that the study is based on road topology, implying the utilization of complex network theory and topological characteristics of the road network. However, the approach employed in this study appears to generate a common topological network, which is commonly used in road network production, particularly in GIS approaches. It is essential to clarify how this paper contributes to the understanding of road topology and what distinguishes the approach used in this study from previous research. Providing these details will enhance the paper's overall contribution.
Response 1: Thanks for your professional suggestion. In addition to most of the characteristics of complex weighted networks, urban traffic networks as spatial networks also have some characteristics different from abstract networks, which determine the topological properties of urban traffic networks. In this paper, we used the primal method to construct the traffic topology network in Xi'an city, which abstracted intersections and intersections as nodes, and the sections connecting two intersections as edges. The original method is the traditional method of real network modeling, which is widely used in various researches. Its modeling process is simple and intuitive, and it can retain relatively complete geographic information about urban networks. Since the cases selected in this paper come from real event information, we hope that the topological network can restore its real geographic information and features to the greatest extent, so that it has clear meaning, rather than the relationship defined in the abstract space. Therefore, we chose a more basic topology structure to maximize the real characteristics of the disaster ontology. We sincerely thank you for your valuable advice. The innovation of road topology networks and related methods will become our next research direction.
Point 2 : The paper lists three causes of water damage: 1) lagging construction of water conservation facilities, 2) increased percentage of impervious area, and 3) special climatic and construction conditions. However, it is unclear how these causes were detected and assessed in this study. Are they assumptions made by the authors or based on statistical data from other research? Linking these causes back to the results of the current study will strengthen the arguments and provide a clearer understanding of their relevance.
Response 2 : We think this is a good suggestion. We have revised section 5.1. We related some of this part to the results of the previous study, while still retaining some general reasons, such as abnormal climate, which we believe is also a significant cause of rainstorm waterlogging.
Point 3 : In section 5.2, the paper presents general governance recommendations. To improve the practicality and effectiveness of these recommendations, it would be beneficial to link them back to the specific results and findings of the study. This linkage will demonstrate how the proposed recommendations address the identified issues and align with the study's outcomes, enhancing the paper's overall impact.
Response 3 : Thank you for your professional advice. It helps us a lot. We have linked them back to the specific results and findings of the study.
Point 4 : The paper would benefit from a clearer statement regarding the main contribution of this research. It is important to explicitly state the unique contributions that this study offers to the field. This clarification will enable readers to better understand the significance and value of the research presented in the paper.
Response 4 : We think this is an excellent suggestion. We have supplemented this part according to the Reviewer's suggestion, where the change can be found in the last paragraph of the revised manuscript.
Point 5 : The abstract is generally well-written. However, a few minor improvements could enhance its clarity and readability. Here are a few suggestions:
- a) Consider rephrasing the sentence "This paper takes the main urban area of Xi'an as the research object" to "This paper focuses on studying the main urban area of Xi'an" or "This paper examines the impact of heavy rainfall on the urban transportation network in Xi'an's main urban area." This helps avoid the use of "research object," which may sound a bit technical.
- b) In the sentence "quantifies and analyzes the topological and geographic structural characteristics of the affected sections due to heavy rainfall," consider clarifying what is meant by "affected sections." Are these sections of the road network that experience damage or disruptions due to heavy rainfall? Providing a brief explanation will improve understanding.
- c) The phrase "impact of heavy rainfall on the functional indexes of the road network" can be made more specific. What specific functional indexes are being considered? For example, you could mention parameters like traffic flow, congestion, travel time, or accessibility.
- d) Consider restructuring the sentence "It is an important reference for improving urban traffic emergency management capability and promoting sustainable urban development." It could be revised as follows: "The findings of this study serve as a valuable reference for enhancing urban traffic emergency management capabilities and fostering sustainable urban development."
Response 5 : Thanks for your professional suggestion. The abstract has been revised.
Thank you very much for your attention and time. Look forward to hearing from you.
Yours sincerely ,
Jiayu Liu, Xiangyu Yang, and Shaobin Ren
29 June 2023

Round 2
Reviewer 2 Report
-